# Real-time surveillance of international SARS-CoV-2 prevalence using systematic traveller arrival screening: An observational study

**Adam J. Kucharski**[1,2]*, **Kiyojiken Chung**[2], **Maite Aubry**[2], **Iotefa Teiti**[2], **Anita Teissier**[2], **Vaea Richard**[2], **Timothy W. Russell**[1], **Raphaëlle Bos**[3], **Sophie Olivier**[3], **Van-Mai Cao-Lormeau**[2]

**1** Centre for Mathematical Modelling of Infectious Diseases, London School of Hygiene & Tropical Medicine, London, United Kingdom, **2** Laboratory of Research on Emerging Viral Diseases, Institut Louis Malardé, Papeete, French Polynesia, **3** Clinical Laboratory, Institut Louis Malardé, Papeete, French Polynesia

* adam.kucharski@lshtm.ac.uk

**Data Availability Statement:** Model code and data are available at: https://github.com/institutlouismalarde/covid-travel-testing.

**Funding:** AJK was supported by a Sir Henry Dale Fellowship jointly funded by the Wellcome Trust

## Abstract

### Background

Effective Coronavirus Disease 2019 (COVID-19) response relies on good knowledge of population infection dynamics, but owing to under-ascertainment and delays in symptom-based reporting, obtaining reliable infection data has typically required large dedicated local population studies. Although many countries implemented Severe Acute Respiratory Syndrome Coronavirus 2 (SARS-CoV-2) testing among travellers, it remains unclear how accurately arrival testing data can capture international patterns of infection, because those arrival testing data were rarely reported systematically, and predeparture testing was often in place as well, leading to nonrepresentative infection status among arrivals.

### Methods and findings

In French Polynesia, testing data were reported systematically with enforced predeparture testing type and timing, making it possible to adjust for nonrepresentative infection status among arrivals. Combining statistical models of polymerase chain reaction (PCR) positivity with data on international travel protocols, we reconstructed estimates of prevalence at departure using only testing data from arrivals. We then applied this estimation approach to the United States of America and France, using data from over 220,000 tests from travellers arriving into French Polynesia between July 2020 and March 2022. We estimated a peak infection prevalence at departure of 2.1% (95% credible interval: 1.7, 2.6%) in France and 1% (95% CrI: 0.63, 1.4%) in the USA in late 2020/early 2021, with prevalence of 4.6% (95% CrI: 3.9, 5.2%) and 4.3% (95% CrI: 3.6, 5%), respectively, estimated for the Omicron BA.1 waves in early 2022. We found that our infection estimates were a leading indicator of later reported case dynamics, as well as being consistent with subsequent observed changes in seroprevalence over time. We did not have linked data on traveller demography or unbiased domestic infection estimates (e.g., from random community infection surveys) in the USA

and the Royal Society (grant Number 206250/Z/17/Z) and the NIHR HPRU in Modelling and Health Economics, a partnership between PHE, Imperial College London and LSHTM (grant code NIHR200908). The views expressed are those of the authors and not necessarily those of the United Kingdom (UK) Department of Health and Social Care, the National Health Service, the National Institute for Health Research (NIHR), or Public Health England (PHE). The funders had no role in study design, data collection and analysis, decision to publish, or preparation of the manuscript.

**Competing interests:** The authors have declared that no competing interests exist.

**Abbreviations:** COVID-19, Coronavirus Disease 2019; CrI, credible interval; ONS, Office for National Statistics; PCR, polymerase chain reaction; REACT-1, Real-Time Assessment of Community Transmission; SARS-CoV-2, Severe Acute Respiratory Syndrome Coronavirus 2.

and France. However, our methodology would allow for the incorporation of prior data from additional sources if available in future.

## Conclusions

As well as elucidating previously unmeasured infection dynamics in these countries, our analysis provides a proof-of-concept for scalable and accurate leading indicator of global infections during future pandemics.

## Author summary

### Why was this study done?

• During the Coronavirus Disease 2019 (COVID-19) pandemic, the true dynamics of infections have been poorly understood globally.
• Although community infection surveys sampling individuals regardless of symptoms have provided crucial insights to inform policy in countries like the United Kingdom, expense and logistical challenges have prevented similar roll-out elsewhere.

### What did the researchers do and find?

• We identified an alternative source of routine information that can provide comparable insights on infection dynamics: Our analysis demonstrates that travel testing data among international arrivals can be used to reconstruct Severe Acute Respiratory Syndrome Coronavirus 2 (SARS-CoV-2) prevalence in multiple countries.
• Applying our method to more than 222,000 arrival tests conducted in French Polynesia between July 2020 and March 2022, we estimated a peak infection prevalence at departure of around 2% in France and 1% in the USA in late 2020/early 2021, with a median prevalence of around 5% and 4%, respectively, estimated for the Omicron BA.1 waves in early 2022.
• We found that our infection estimates were a leading indicator of the later observed case dynamics in these countries and were consistent with subsequent observed changes in seroprevalence over time in France and the USA.

### What do these findings mean?

• Our results suggest that systematic collection of traveller testing data can enable real-time estimation of underlying epidemic dynamics in multiple countries.
• In our study, personal data about travellers—such as age and address—was not available for analysis. In future, linking traveller tests to demographic characteristics most relevant to infection status could enable a more detailed understanding of risk.

## Introduction

Understanding the true extent of Severe Acute Respiratory Syndrome Coronavirus 2 (SARS-CoV-2) infection within a population is crucial for effective planning and response. As well as being a leading indicator of subsequent disease, it can help inform the timing and design of

control measures both domestically and with respect to internal travel [1]. However, throughout the pandemic, insights into underlying infection dynamics have typically been limited, hindering countries' ability to respond promptly and proportionately.

In the early stages of 2020, large undetected epidemics in several locations were first identified as a result of infected travellers from these areas being detected in other countries [2,3]. Limited testing availability also meant that infection dynamics had to be estimated from lagged indicators such as hospitalisations, with the number of infections derived from available data on severity [4]. Cross-sectional seroprevalence studies have since provided estimates of the extent of infection within populations, but with a lag and without information on when those infections occurred [5]. Moreover, the roll-out of vaccination and emergence of novel variants means estimation of infection dynamics based on severe outcomes or serological data is becoming more challenging [6]. In some instances, countries have tackled these issues by setting up routine community sampling regardless of symptoms, such as the ONS (Office for National Statistics) community infection survey and REACT-1 (Real-Time Assessment of Community Transmission) in the United Kingdom [7,8], as well as cohort studies tracking infection in specific workplaces, such as healthcare workers [9]. However, the expense and logistical complexity of such studies have resulted in limited deployment globally.

Despite a lack of studies tracking local infection prevalence, large numbers have been tested regardless of symptoms during the pandemic as a result of traveller screening programmes [10]. Countries have predominately used travel testing as a method of control, with positive individuals having to isolate, as well as being prevented from travelling at all if they test positive before departure. Algorithms have also been developed to predict individuals more likely to test at arrival based on demographic characteristics such as age and country of origin [11], which can be useful for detecting infections, but less relevant if the objectives to obtain unbiased estimates of overall population prevalence.

Although some countries with strict restrictions on traveller numbers have reported the number of arriving infections detected over time [12] and data have been reported from brief or small-scale testing programmes [13–15], there has been no systematic long-term data published from global travel testing. Tests have been conducted by different companies and agencies, and in combinations that can include both departure and arrival testing, typically without collation of these tests in consistent databases.

Despite the inconsistent way in which travel testing has generally been reported during the pandemic, such screening presents a unique opportunity for real-time surveillance of infection in multiple countries. Local community infection studies, or analysis of departure testing results, can only provide information on infections within the country conducting the study. In contrast, arrival testing can provide insights into infections among travellers from a range of different locations. However, there are challenges to interpreting arrival test results. If testing is implemented at both departure and arrival, then infections detected among arriving travellers only reflect a subset of all the infected individuals who attempted to travel.

Using data on how test positivity varies over the course of infection, and details of departure and arrival protocols, we developed a model of the travel screening process, and hence used arrival prevalence to reconstruct how many travellers would have tested positive in countries of departure. Incorporating data from French Polynesia, which had systematic arrival screening for SARS-CoV-2, we then test the potential to use local surveillance to recover underlying international infection dynamics.

## Methods

### Ethics statement

Communication of data from the surveillance system was approved by Comité d'Ethique de la Polynésie française (ref Avis n°90 CEPF 15_06_2021). Secondary data analysis was approved by the London School of Hygiene & Tropical Medicine Observational Research Ethics Committee (ref 28129).

### Data

Travel testing in French Polynesia was conducted in 2 distinct phases, with some additional adjustments to protocols during each phase. The first phase, between 15th July 2020 and 30th April 2021, was the "COV-CHECK" system, with travellers required to perform a polymerase chain reaction (PCR) test less than 72 h before departure as well as a self-test 4 days after arrival in French Polynesia [16]. Mandatory quarantine was added to this protocol on 20th February 2021. A schematic of the process is shown in Fig 1A.

The second phase started on 1st May 2021 with the option of taking either an antigen test within 48 h of departure or a PCR test within 72 h. Moreover, testing was performed on the day of arrival in French Polynesia [17] by nurses until 12th August 2021, then using the self-test COV-CHECK protocol until 27th December 2021, and finally reverted to nurses until the end of the surveillance period. Unvaccinated individuals (mostly children) had additional self-tests on day 4 and day 8 until 19th January 2022, then on days 2 and 5. As these repeat tests did not systematically have accompanying dates in the dataset, and hence, it was not possible to identify which test was conducted on day 0, we omitted these repeated test results (7,809 of 112,945 total test results that had a recorded country of origin) from the analysis. After 28th December 2021, departing travellers could take either an antigen test within 24 h of travel or a PCR test within 72 h. Throughout the pandemic, all arrival PCR tests were screened for mutations associated with novel variants and subset were sequenced to identify the specific imported variants, with sequencing activity typically focused in the early stages of each wave to detect initial arriving infections.

To protect privacy, age, nationality, and address of tested passengers were not available for analysis. The origin of tested passengers was therefore defined based on the flight number. Between 15th July 2020 and 17th November 2021, passengers on flights from Paris, France had a direct transit (via Canada or Guadeloupe) so it was not possible to leave or join the plane in transit. Later, this transit was via the United States of America (Los Angeles), where it was possible to leave or join the plane mid-journey. Between July 2020 and March 2022, there were 464,728 incoming travellers: 229,744 tourists and 234,984 returning residents [18]. We do not know whether passengers that took their plane in Paris or in Los Angeles came originally from another country, but because 90% of tourist arrivals in 2021 were reported as from these 2 original countries on arrival forms, we made the assumption that infected tourists or returning residents reflect infections acquired in the USA and France.

The primary public health aims of travelling testing in French Polynesia were 2-fold: (1) to limit virus dissemination (travellers found positive for SARS-CoV-2 were asked to self-isolate and were not allowed to travel to outer islands); and (2) to obtain early information on the introduction of new variants circulating in other countries and to be prepared for local circulation. The cost of traveller surveillance in French Polynesia was initially covered by the local government (until July 2021), then by the travellers themselves when registering on the (mandatory) travellers identification platform.

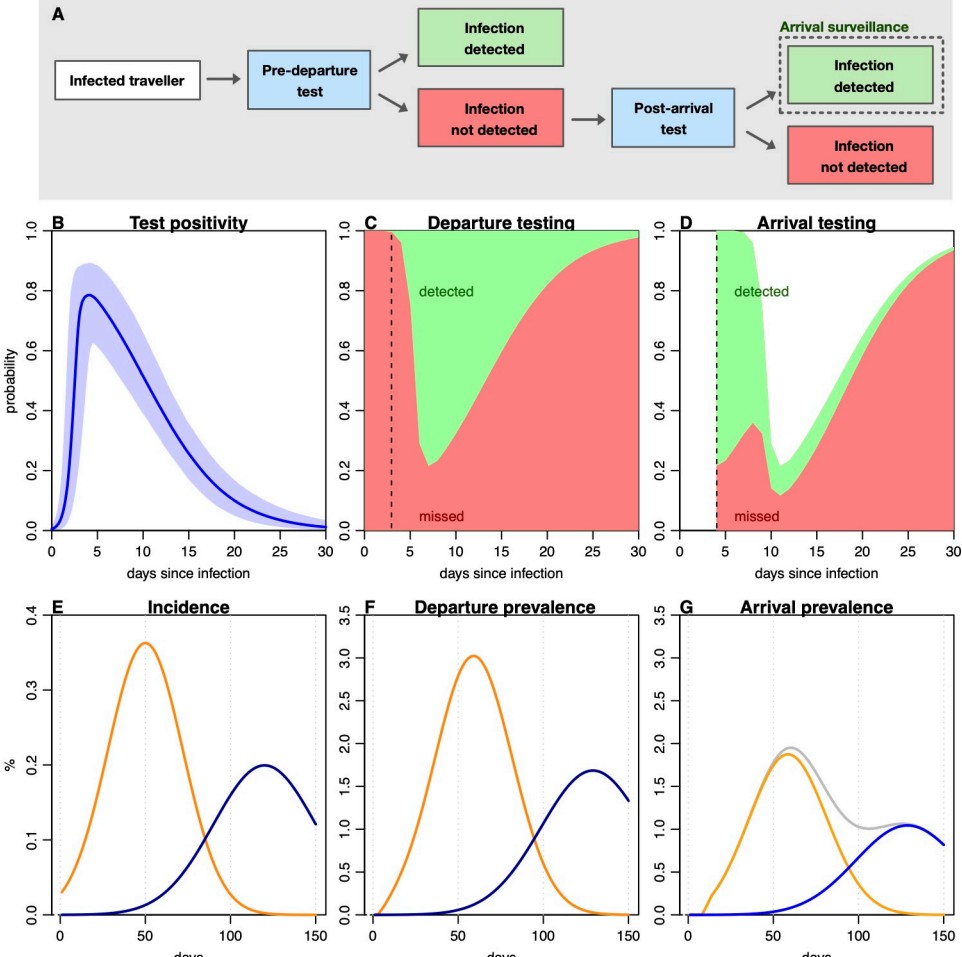

**Fig 1. Impact of departure and arrival testing protocols on PCR prevalence.** (A) Possible outcomes for infected travellers in a scenario with predeparture and post-arrival testing. (B) Probability an individual will test PCR positive at different points since infection, based on self-tested asymptomatically tested participants [18]. Line shows median, with shaded region showing 95% CrI in this Bayesian analysis. (C) Probability infected traveller will be detected by a predeparture test, in scenario where test conducted 2 days before departure (dashed line). (D) Probability infected traveller will be detected by a post-arrival test conducted 4 days after arrival (dashed line), assuming no local acquisition of infection. (E) Illustrative epidemic showing proportion of population newly infected per day with 2 different variants. (F) Larger measured prevalence in predeparture testing corresponding to incidence curves in (E), based on positivity probability in (C). (G) Measured prevalence in post-arrival testing, corresponding to incidence curves in (E), based on positivity probability in (D). Grey line shows cumulative prevalence. CrI; credible interval; PCR, polymerase chain reaction.

## Estimation of departure prevalence from arrival testing

If testing is implemented at both departure and arrival, then infections detected among arriving travellers only reflect a subset of all the infected individuals who attempted to travel (Fig 1A). Analysis of repeated PCR self-testing in a healthcare worker cohort [19] found that individuals can test positive for several weeks, with a peak in detection around 5 days post infection (Fig 1B). Departure testing is therefore most likely to prevent the most detectable infections from travelling (Fig 1C), which means if arrival testing is conducted shortly after arrival, the proportion of infections detected will be distributed differently to the departure tests (Fig 1D). For a given incidence of new infections in a country of departure, measured

prevalence among departing and arriving travellers could therefore be considerably different, with overall arrival prevalence masking underlying dynamics in countries of departure (Fig 1E–1G).

We can estimate departure prevalence from arrival testing data as follows. If $N$ individuals are tested at arrival, then the number who test positive is binomially distributed with probability q, where q = P(test positive at arrival | tested negative at departure) = P(arrival+ | departure–).

We can rewrite this probability in terms of arrival and departure tests:

$$P(arrival + |departure-) = P(arrival + \text{ and departure}-)/P(departure-).$$

If w is the probability an individual is infected at departure (i.e., could in theory test positive), then:

$$P(arrival + \text{ and departure}-) = w\, P(arrival + \text{ and departure} - | \text{ infected})$$

$$P(departure-) = w\, P(departure - | \text{ infected}) + (1 - w).$$

Details of how these probabilities can be calculated are provided in S1 Text.

Combining the above probabilities gives the following expression for the probability an individual tests positive at arrival:

$$q = \frac{w\, P(arrival + \text{ and departure} - | \text{ infected})}{w\, P(departure - | \text{ infected}) + (1 - w)}.$$

If we have observations on the number tested at arrival, and the number tested, we can therefore estimate the distribution of w, and hence, prevalence at departure (full details in S1 Text). Model code and data are available at: https://github.com/institutlouismalarde/covid-travel-testing.

## Pooling test model

For a pool of size $n$, the probability that there is at least 1 positive in the pool given true prevalence $x$ is equal to p = 1-(1–$x$)$^n$. For $N$ pools, the likelihood of x is therefore given by the binomial distribution B(N,p). For each scenario, we calculated the maximum likelihood estimate for x and corresponding confidence interval generated from the profile likelihood.

## Results

The effectiveness of traveller screening based on biological outcomes, such as symptoms or test positivity, will depend on how long ago travellers were infected. In turn, this will depend on the dynamics of the epidemic in countries of departure [20]. During a growing epidemic, individuals are more likely to have been infected recently, whereas infections will generally be older in a declining epidemic. In situations where test positivity is influenced by the phase of the epidemic, it is often necessary to reconstruct infection dynamics using computationally intensive latent processes models [21]. However, we found that the presence of departure testing changes the distribution of infection timings among arrivals, as the individuals most likely to be detectable are least likely to travel (Fig 2A–2C). As a result, most arriving infections will have been infected within a narrower window that those detected at departure. This reduced the impact of epidemic phase on arrival positivity and allowed for reconstruction of departure prevalence from arrival data with a single multiplier calculated from PCR positivity curves and the specific departure and arrival testing protocol. Our approach could therefore generate

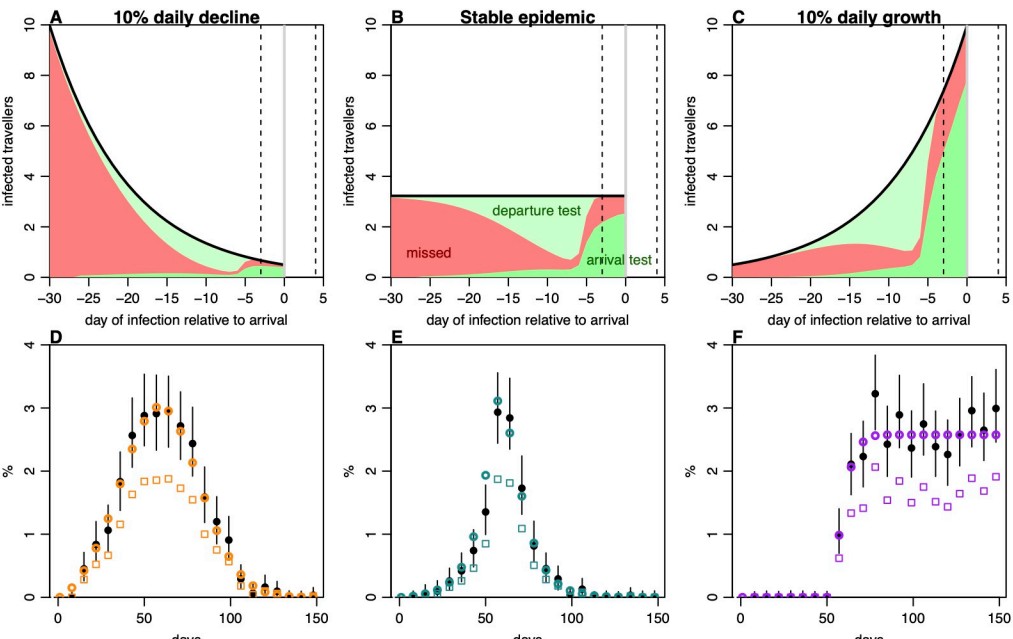

**Fig 2. Observation and estimation of infection prevalence under different epidemic dynamics.** (A) Proportion of infected travellers detected relative to infection time, in a scenario with incidence declining 10% per day, and testing 2 days before departure, with 1 day in transit, and 4 days after arrival (dashed lines). Light green, travellers detected predeparture; dark green, travellers detected post-arrival; red, travellers missed. (B) Scenario in a stable epidemic, i.e., 0% daily change in incidence. (C) Scenario in epidemic with 10% daily growth. (D) Reconstruction of simulated epidemics from arrival testing data, assuming 5,000 arrivals tested per week. Solid points, "true" prevalence at departure with lines showing 95% binomial confidence interval; squares, measured prevalence at arrival in simulated scenario with a test 2 days before departure with 1 day in transit and another test 4 days post-arrival, with circles showing reconstructed departure prevalence from these data. (E, F) Reconstruction as described in (D) under different assumed epidemic dynamics.

accurate estimates for simulated departure prevalence dynamics, particularly if there was a gap of several days between the departure and arrival test, which reduced the variance of the time-since-infection for arrivals (Fig 2D–2F). In a sensitivity analysis, we found that estimates were more reliable with larger sample sizes and less biased when the testing process was accurately quantified, with correct assumptions about delay between tests and test sensitivity (S1 Fig).

To show how this approach can be used mid-pandemic, we analysed traveller data collected as part of the Coronavirus Disease 2019 (COVID-19) response in French Polynesia [16]. Between July 2020 and March 2022, more than 222,000 traveller tests were conducted as part of an arrival screening programme (Fig 3A). During the first phase of the surveillance strategy, travellers performed a self-test (COV-CHECK protocol) 4 days after arrival as well as a PCR test within 72 h of departure. Faced with novel variants, mandatory quarantine was added to the protocol in February 2021. Between May 2021 and March 2022, testing was performed on the day of arrival first by nurses until 12th August 2021, then using the self-test COV-CHECK protocol until 27th December 2021, and again by nurses until the end of the surveillance period. During 2021, the option was also added for an antigen test within 48 h of departure rather than a PCR test within 72 h.

There were 1,341 positive arrival tests in the dataset analysed, with considerable variation in weekly prevalence over the course of the pandemic (Fig 3B). There were 3 main COVID-19 waves in French Polynesia—caused by wild type, Delta and Omicron—with Alpha detected among travellers without leading to widespread local transmission (Fig 3C). Among arriving

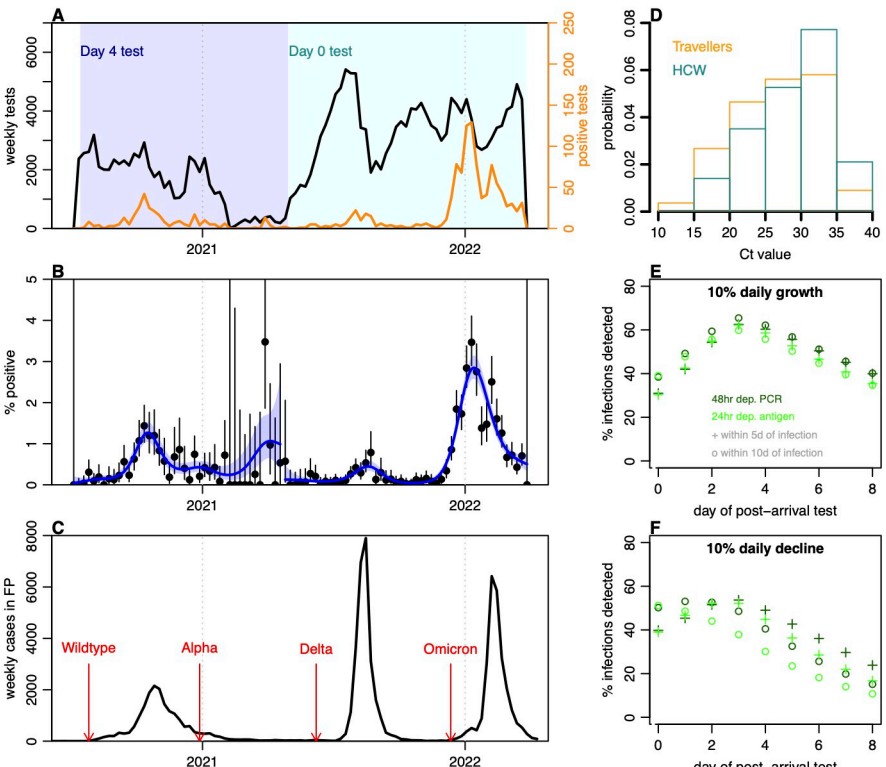

**Fig 3. Arrival testing in French Polynesia, July 2020 to March 2022.** (A) Testing data and changes in main protocols over time. Initially travellers were tested 4 days post-arrival (d4), with additional quarantine introduced on 20th February 2021; later vaccinated travellers were tested on day of arrival (d0), with additional testing on day 4 and 8 for non-vaccinated individuals. Black line, number of tests performed; orange line, number of positive tests. (B) Percent of tests that were positive over time, with lines showing binomial confidence interval and blue line showing GAM fit with shaded 95% CI. (C) Local COVID-19 cases reported in French Polynesia, with arrows showing first detection of different variants among travellers. (D) Distribution of Ct values among positive arrivals into French Polynesia (orange bars) and routinely tested UK HCWs (cyan bars). (E) Estimated percent of infections that would be detected early on (i.e., within 5 or 10 days of infection) under different travel testing protocols using PCR and antigen tests in a growing epidemic. (F) Estimated percent of infections detected in a declining epidemic. COVID-19, Coronavirus Disease 2019; Ct, cycle threshold; GAM, generalised additive model; HCW, healthcare worker; PCR, polymerase chain reaction.

travellers, 15% of measured cycle threshold (Ct) values were below 20 (Fig 3D), with a distribution that was lower than has previously been observed in self-testing of healthcare workers [19]. This is to be expected given travellers will typically be earlier in their infection at the point of the arrival test, as noted above (Fig 2). Such a pattern is also consistent with previous reports of shedding being higher than usual among positive arriving travellers [22]. Based on protocols implemented, we estimated that arrival PCR testing at day 4 would have been expected to detect around 60% of infected individuals within the first 10 days of their infection if an epidemic in the departure location was growing at 10% per day (Fig 3E), and around 40% during an epidemic that was declining 10% per day (Fig 3F). This illustrates the value of delayed arrival testing for detecting infections during rising SARS-CoV-2 transmission, as well as for reconstructing departure prevalence. However, given the percentages undetected, travel testing and quarantine would still need to be combined with other measures, such as a reduction in traveller numbers or domestic control measures, to prevent sustained transmission. In reality, the proportion of infections detected would be binomially distributed, and hence also subject to randomness, as well as from any uncertainty in the test sensitivity at departure and arrival.

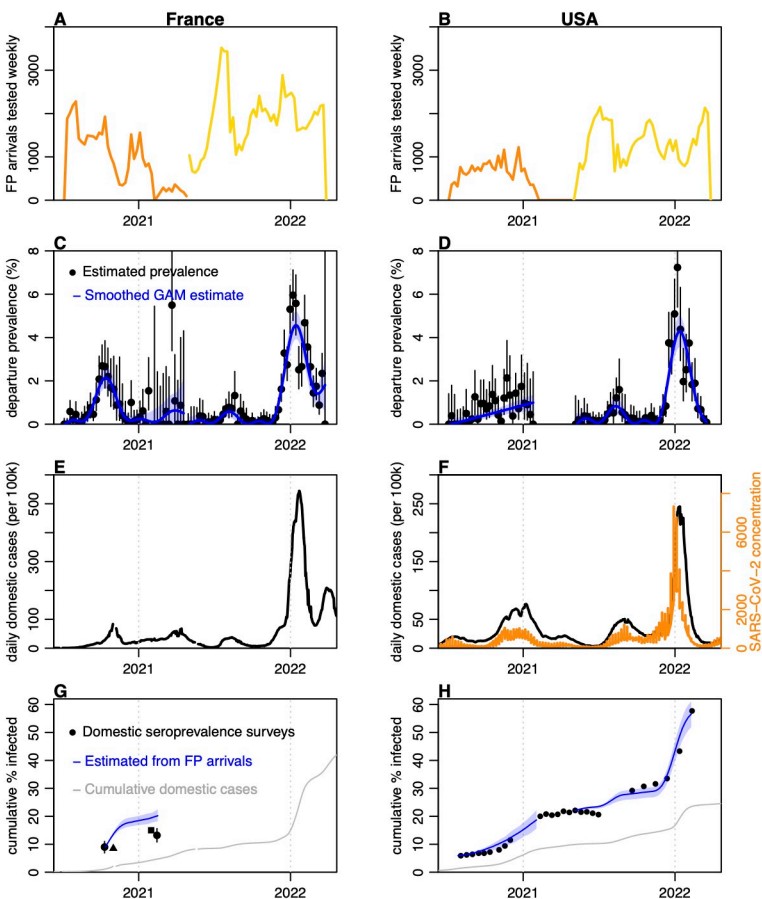

**Fig 4. Reconstruction of infection dynamics in France and USA from arrival testing data in French Polynesia.** (A) Number of arrivals from France tested per week. Orange, tests performed at day 4 after arrival; yellow, tests performed at day of arrival. (B) Arrival testing from USA. (C) Estimated prevalence among arrivals from France, with maximum a posteriori estimate shown by black dots with lines showing Bayesian 95% high posterior density interval; blue line and shaded region, GAM fit to these data and 95% prediction interval. (D) Estimated prevalence among arrivals from the USA. (E) Domestic cases reported in France. (F) Domestic cases in USA, shown by black line, alongside SARS-CoV-2 concentration in wastewater [27], shown by orange line. (G) Comparison of estimated cumulative infections and observed seroprevalence in France. Black dots, observed national seroprevalence in France in October 2020 and February 2021 [28]; black triangle, observed national seroprevalence in November 2020 [29]; black square, estimated proportion infected by January 2021 [30]; blue line, cumulative incidence derived from blue line in (E), shifted to match initial value of black line; shaded region, bootstrap 95% prediction interval; grey line, cumulative per capita domestic cases reported. (H) Estimated cumulative infections and observed seroprevalence in USA. Black dots and lines, observed seroprevalence over time with 95% confidence interval [5]; red lines, estimated cumulative incidence over same periods, shifted to match initial values. GAM, generalised additive model; SARS-CoV-2, Severe Acute Respiratory Syndrome Coronavirus 2.

The majority of tested arrivals into French Polynesia had either the USA or metropolitan France (with a few hours transit via the USA or Canada or Guadeloupe) as the origin for their trip (Fig 4A and 4B). This was consistent with reported residency data on immigration forms, with 90% of tourist arrivals in 2021 from these 2 countries [18]. We found that estimate prevalence at departure among travellers from the USA or France (Fig 4C and 4D), based on French Polynesia arrival tests, anticipated observed case dynamics in the 2 countries (Fig 4E and 4F), showing the value of traveller testing as a leading indicator. Peaks in prevalence occurred shortly before peaks in reported cases, as would be expected due to delays in symptom onset, symptomatic testing, and reporting in the country of origin. Adjusting for traveller testing

protocols as outlined in Fig 1, we estimated a peak infection prevalence at departure of 2.1% (95% credible interval (CrI): 1.7, 2.6%) in France and 1% (95% CrI: 0.63, 1.4%) in the USA in late 2020/early 2021, with prevalence of 4.6% (95% CrI: 3.9, 5.2%) and 4.3% (95% CrI: 3.6, 5%), respectively, estimated for the Omicron BA.1 waves in early 2022.

To further assess the ability of arrival screening to estimate international SARS-CoV-2 dynamics, we converted prevalence at departure (i.e., current positivity) into an estimate of incidence (i.e., new daily infections) and compared cumulative incidence over the same period as repeated serological surveys in France and the USA. In France, our estimate of cumulative incidence in late 2020 was slightly larger than implied in serological studies (Fig 4G). There are several potential explanations for this discordance. The volume of travel from France declined substantially during that wave, which may have affected representativeness of travellers, as risk-averse individuals could have been less likely to travel, and risk-taking individuals more likely to be infected in transit. As testing was conducted on day 4 during this period and there was a COVID-19 wave in French Polynesia in late 2020, there is also the possibility of infection post-arrival. In contrast, we found that our estimates for the USA closely matched increases in seroprevalence during epidemic waves in both late 2020 and late 2021 (Fig 4H).

Because systematic individual-level testing is resource intensive, we also explored the potential for pooled testing to generate estimates of prevalence at arrival. For example, travellers from a given airport of origin could submit a swab via a pooled collection system, with testing returning either a positive or negative result for each pool. We estimated that for a sample of 100 incoming travellers, there was considerable uncertainty in prevalence estimates when larger pool sizes were used (Fig 5A). However, this uncertainty was reduced for larger traveller volumes (Fig 5B). Surveillance design (e.g., pool size and whether aggregated daily or weekly) could therefore be tailored to ensure that prevalence estimates for a given pathogen provide the desired level of precision and/or required degree of confidence that prevalence is below a certain value (Fig 5C and 5D).

## Discussion

Our analysis shows that it is possible to obtain detailed international epidemiological insights from arrival testing protocols that were designed predominantly as control measure to reduce risk of onwards local transmission. As a proof-of-concept, we reconstructed epidemic dynamics in France and the United States using arrival data from French Polynesia between July 2020 and March 2022.

Raw observed infection prevalence at arrival in French Polynesia was consistent with broad ranges observed in other, smaller-scale studies. Among arrivals into Alaska from June to November 2020, 0.8% were positive [13]. Infection prevalence at arrival was 1.0% in Toronto, Canada during September and October 2020 [15] and 1.5% in Alberta, Canada in November 2020 [14]. However, travellers in these studies did not have their country of origin reported, or information on departure testing protocol, limiting the ability of these datasets to provide comparable information on likely prevalence in countries of departure. Our estimates of peak prevalence in France and the USA were also similar in magnitude to PCR test positivity in the ONS community infection survey in the UK, which peaked at around 2% during the Alpha wave in early 2021 and around 7% during the BA.1 wave in late 2021/early 2022 [7].

A key strength of our analysis is that it uses data from a systematic testing programme with known protocols and countries of departure. Despite the relatively small travel volume into French Polynesia, our real-time estimates of cumulative incidence in France and the USA reflected subsequent changes in national seroprevalence observed in these locations. In contrast, commonly cited international COVID-19 indicators such as cases or hospitalisations are

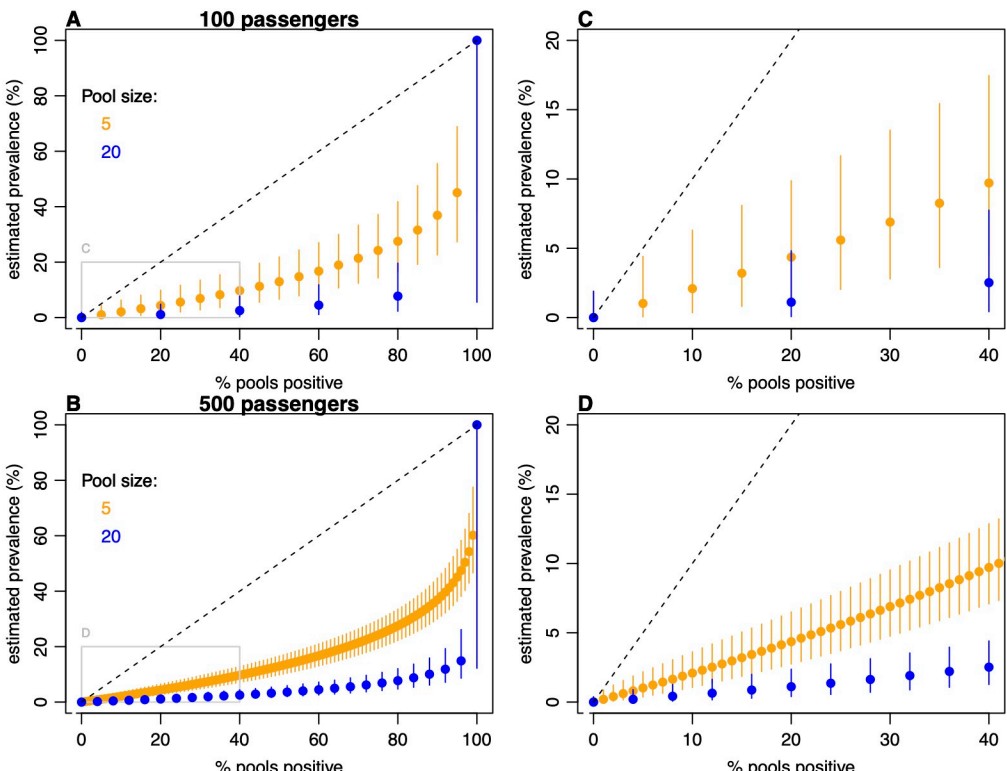

**Fig 5. Estimation of prevalence based on pooled testing.** (A) Scenario where 100 passengers are tested, either in pools of size 5 or 20, with the proportion of positive pools used to estimate prevalence. Dots show mean estimate, with 95% confidence interval shown by lines. Dashed line shows equivalent calculation with individual-level testing (i.e., pool of size 1). (B) Scenario where 500 passengers are tested. (C, D) close-up of the boxed region in (A) and (B), focusing on the estimation range covering the prevalence values observed for SARS-CoV-2 in Fig 4. SARS-CoV-2, Severe Acute Respiratory Syndrome Coronavirus 2.

strongly dependent on symptom severity and domestic access to testing, as well as being delayed outcomes relative to infection incidence. Although symptomatic individuals may have avoided travel, test positivity also tends to peak around the time of symptom onset [23], and hence, these individuals would have been detected at departure had they travelled, a process that is already accounted for in the analysis presented here.

In French Polynesia, further data linkage was limited to protect traveller privacy, meaning it was not possible to explore how factors such as age or occupation influenced prevalence. Although our estimates reflected national-level patterns of seroprevalence in the USA and France, the age distribution of travellers may be different to the age distribution of the population at departure, particularly if there are additional constraints on unvaccinated individuals such as children travelling, as occurred from mid-2021 until early 2022. Moreover, travellers may be from wealthier or less-risk averse groups, who may have a different exposure risk to the wider population. Demographic insights could be expanded in future epidemics by routinely reporting data on epidemiologically relevant values such as age and occupation, which would allow adjustment for potential biases when estimating prevalence in countries of departure. Linkage with test results at departure would also enable validation of assumptions about PCR sensitivity at each stage of travel.

There are some additional limitations to our proof-of-concept analysis. The testing protocols in French Polynesia enabled estimation of departure prevalence using a binomial

distribution, with little bias from epidemic phase (Fig 2E–2G). In the absence of departure testing, such methods would no longer be reliable, and arrival testing would need to be analysed in the context of changing epidemic dynamics, requiring more complex inference methods that also model the underlying epidemic process [21]. We also assume that PCR positivity over time for the wild-type variant is representative of subsequent variants, based on similarities observed in cohort studies [23,24]. If positivity were to vary substantially, then this would need to be accounted for with different adjustments based on dominant variants in countries of departure. If viral shedding profiles were very different by variant, similar adjustments would also need to be made when inferring infection dynamics from other data sources, such as community infection surveys or wastewater data.

During the initial phase of COVID-19 pandemic, testing among departing and arriving travellers provided valuable indications of the true extent of infection [3], which were in turn used to estimate crucial metrics such as infection fatality risk [25]. Understanding levels of community infection has been challenging both when infections are rising, with the epidemic outstripping symptomatic surveillance capacity, as well as in the later stages of the COVID-19 pandemic, with countries rolling back test availability for symptomatic cases. In the UK, studies such as REACT-1 and ONS have provided continuity in local understanding of community infections, but such data streams have been extremely rare globally, given their expense and logistical complexity. However, our analysis shows that routine traveller testing can enable similar insights at an international scale, using protocols that were already being implemented in many countries, offering a substantial improvement on common situational awareness based on reported cases, hospitalisations, or deaths [4]. Such estimates could also be triangulated against other potential leading indicators, including viral concentrations in wastewater, although interpretation of wastewater data would rely on additional analysis to convert into a comparable measure of the proportion of the population who are currently infected. In turn, these measurements of infection dynamics could be used in estimation of situation awareness metrics such as the reproduction number, as well as estimation of population-level epidemic dynamics for scenario models, where knowledge of the extent of infection can substantially improve statistical inference [26].

Our study provides a proof-of-concept for ongoing COVID-19 management and future pandemic planning, showing that systematic collection of testing data with minimal linkage can enable real-time estimation of underlying epidemic dynamics in multiple countries. Moreover, deployment of such methods in multiple locations—such as key global travel hubs— would allow for synthesis of prevalence estimation across datasets, narrowing uncertainty, and greatly expanding the network of countries covered.

## Supporting information

**S1 Text. Supplementary methods.**
(DOCX)

**S1 Fig. Sensitivity of estimation to difference sample sizes and testing assumptions.** (A) Reconstruction of simulated epidemics from arrival testing data, assuming 100 arrivals tested per week. Solid points, "true" prevalence at departure with lines showing 95% binomial confidence interval; squares, measured prevalence at arrival in simulated scenario with a test 2 days before departure with 1 day in transit and another test 4 days post-arrival (i.e., 2d + 4d), with circles showing reconstructed departure prevalence from these data. (B) Same as (A), but with 5,000 arrivals tested per week. (C) Same as (A), but with 20,000 arrivals tested per week. (D) Reconstruction of simulated epidemics from arrival testing data when test timing is mis-specified (shown in green). In the simulation, there is a test 2 days before departure with 1 day in

transit and another test on day of arrival. However, inference is performed with a test 2 days before departure with 1 day in transit and another test 4 days post-arrival. (E) Same as (D), but with 5,000 tested per week. (F) Same as (D), but with 20,000 tested per week. (G) Reconstruction of simulated epidemics from arrival testing data when test sensitivity is mis-specified (shown in purple). Simulated tests have peak sensitivity equal to 80% of the peak in our baseline analysis, with inference performed under assumption sensitivity is unchanged. Otherwise, the scenario is same as in (A). (H) Same as (G), but with 5,000 tested per week. (I) Same as (G), but with 20,000 tested per week.
(TIFF)

**S2 Fig. Reconstruction of infection dynamics in France and USA from arrival testing data in French Polynesia (FP), assuming antigen tests at departure after May 2021.** Panels otherwise as described in main text (Fig 4).
(TIFF)

**S3 Fig. Reconstruction of cumulative incidence under different estimated methods.** (A) Simulated daily incidence. (B) Simulated daily prevalence based on the convolution of incidence in (A) and median probability of PCR positivity in Fig 1B. (C) Reconstruction of incidence from prevalence in (B) using our baseline scaling approximation (red line) as well as deconvolution using the Moore–Penrose generalized inverse (blue line, showing numerical instability at end of time series). (D) Comparison of simulated cumulative incidence (black dots), with estimated cumulative incidence using our baseline scaling approximation (red line) as well as deconvolution using the Moore–Penrose generalized inverse (blue line).
(TIFF)

## Author Contributions

**Conceptualization:** Adam J. Kucharski, Van-Mai Cao-Lormeau.

**Data curation:** Kiyojiken Chung, Iotefa Teiti, Anita Teissier, Vaea Richard, Raphaëlle Bos, Sophie Olivier.

**Formal analysis:** Adam J. Kucharski, Anita Teissier, Vaea Richard, Timothy W. Russell, Raphaëlle Bos, Sophie Olivier.

**Investigation:** Adam J. Kucharski, Kiyojiken Chung, Maite Aubry, Iotefa Teiti, Van-Mai Cao-Lormeau.

**Methodology:** Adam J. Kucharski, Maite Aubry, Timothy W. Russell, Van-Mai Cao-Lormeau.

**Supervision:** Van-Mai Cao-Lormeau.

**Writing – original draft:** Adam J. Kucharski, Van-Mai Cao-Lormeau.

**Writing – review & editing:** Kiyojiken Chung, Maite Aubry, Iotefa Teiti, Anita Teissier, Vaea Richard, Timothy W. Russell, Raphaëlle Bos, Sophie Olivier.

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
