## [Editor Report · Decision Letter 0]

9 Jan 2023

Dear Dr Kucharski, 

Thank you for submitting your manuscript entitled "Real-time surveillance of international SARS-CoV-2 prevalence using systematic traveller arrival screening" for consideration by PLOS Medicine.

Your manuscript has now been evaluated by the PLOS Medicine editorial staff as well as by an academic editor with relevant expertise and I am writing to let you know that we would like to send your submission out for external peer review.

Please re-submit your manuscript within two working days, i.e. by Jan 11 2023 11:59PM.

Kind regards,

Philippa Dodd, MBBS MRCP PhD

PLOS Medicine

---

## [Decision Letter · Decision Letter 1]

26 Apr 2023

Dear Dr. Kucharski,

Thank you very much for submitting your manuscript "Real-time surveillance of international SARS-CoV-2 prevalence using systematic traveller arrival screening" (PMEDICINE-D-22-04017R1) for consideration at PLOS Medicine. 

[LINK]

In light of these reviews, I am afraid that we will not be able to accept the manuscript for publication in the journal in its current form, but we would like to consider a revised version that addresses the reviewers' and editors' comments. Obviously we cannot make any decision about publication until we have seen the revised manuscript and your response, and we plan to seek re-review by one or more of the reviewers. 

We expect to receive your revised manuscript by May 17 2023 11:59PM. Please email us (plosmedicine@plos.org) if you have any questions or concerns.

We look forward to receiving your revised manuscript. 

Sincerely,

Philippa Dodd, MBBS MRCP PhD

PLOS Medicine

plosmedicine.org

GENERAL

Please respond to all editor and reviewer comments detailed below in full.

Please include line (and page) numbers starting at 1 and in continuous sequence thereafter.

COMMENTS FROM THE ACADEMIC EDITOR

Thanks for your submission and congratulations on proposing a potentially unique case for traveller screening data. Please find below some requests:

(1) Most countries are using environmental surveillance (i.e. wastewater surveillance) as a leading indicator. Would suggest including that triangulation and comparison as part of this submission, if feasible.

(2) Please elaborate on utility of these data for other countries and the larger public health community (e.g. would you propose the UK used these data as a proxy for other countries' transmission levels?).

(3) Please elaborate on potential selection biases involved with using traveller data. A couple examples include that these travellers may only represent the wealthy from US/France rather than travellers across socioeconomic strata and also misses symptomatic individuals.

(4) For the French and US analyses, could you contextualize with serosurveillance and case surveillance data from those countries rather than the UK (the results cites UK infection surveys after the analyses which was confusing)?

(5) There is a lot going on in the figures. I really wanted to hone in on the triangulation of travellers data to nationally representative seroprevalence data. This appears to be in C and D? Are the readers supposed to take away that the trend is the same using both data sources? Furthermore, for E and F are these simply data from other studies? It is not immediately clear how these travellers data are included in those panels

TITLE

Please revise your title according to PLOS Medicine's style. Your title must be nondeclarative and not a question. It should begin with main concept if possible. "Effect of" should be used only if causality can be inferred, i.e., for an RCT. Please place the study design ("A randomized controlled trial," "A retrospective study," "A modelling study," etc.) in the subtitle (ie, after a colon).

ABSTRACT

Abstract Background: 

Please ensure that the final sentence clearly states the study question.

Abstract Methods and Findings:

Please ensure that all numbers presented in the abstract are present and identical to numbers presented in the main manuscript text.

Please define ‘PCR’ at first use for the reader

Please provide brief demographic details of the study population (e.g. sex, age, ethnicity, etc)

Please include/clearly define the number of participants, length of follow up, and main outcome measures.

Please define the numerical values contained within parentheses 

Please quantify the main results with 95% CIs and p values. When reporting p values, please report as p<0.001 and where higher the exact p value as p=0.002, for example. If not reporting p values, please clearly state the reasons why not, to help facilitate transparent data reporting.*

When reporting 95% CIs suggest the use of commas to separate upper and lower bounds as opposed to hyphens as these can be confused with reporting of negative values.* 

*Please check and amend throughout the main manuscript, tables, figures and supporting files where relevant. 

Please include any important dependent variables that are adjusted for in the analyses.

Please include numerators and denominators used to derive percentages.

In the last sentence of the Abstract Methods and Findings section, please detail 2-3 primary limitations of the study's methodology.

Abstract Conclusions:

Please emphasize what is new and address the implications of your study and in doing so please avoid assertions of primacy (‘We report for the first time...’), ‘In this study, we observed...’ may be useful.

Please interpret the study based on the results presented in the abstract, emphasizing what is new without overstating your conclusions.

Please address the study implications without overreaching what can be concluded from the data 

Please avoid vague statements such as ‘these results have major implications for policy/clinical care’. Mention only specific implications substantiated by the results.

AUTHOR SUMMARY

At this stage, we ask that you include a short, non-technical Author Summary of your research to make findings accessible to a wide audience that includes both scientists and non-scientists. The authors summary should consist of 2-3 succinct bullet points under each of the following headings:

• Why Was This Study Done? Authors should reflect on what was known about the topic before the research was published and why the research was needed.

• What Did the Researchers Do and Find? Authors should briefly describe the study design that was used and the study’s major findings. Do include the headline numbers from the study, such as the sample size and key findings. 

• What Do These Findings Mean? Authors should reflect on the new knowledge generated by the research and the implications for practice, research, policy, or public health. Authors should also consider how the interpretation of the study’s findings may be affected by the study limitations. In the final bullet point of ‘What Do These Findings Mean?’, please describe the main limitations of the study in non-technical language.

The Author Summary should immediately follow the Abstract in your revised manuscript. This text is subject to editorial change and should be distinct from the scientific abstract. Please see our author guidelines for more information: https://journals.plos.org/plosmedicine/s/revising-your-manuscript#loc-author-summary

INTRODUCTION

Please indicate whether your study is novel and how you determined that. 

If there has been a systematic review of the evidence related to your study (or you have conducted one), please refer to and reference that review and indicate whether it supports the need for your study.

Please define ‘ONS’ and ‘REACT-1’

METHODS and RESULTS

We ask all authors of modelling studies to ensure the inclusion of specific items, derived from Geoffrey P Garnett, Simon Cousens, Timothy B Hallett, Richard Steketee, Neff Walker. Mathematical models in the evaluation of health programmes. (2011) Lancet DOI:10.1016/S0140-6736(10)61505-X. Please review the list below and ensure that all items are included in the relevant parts of the main manuscript:

* Please provide a diagram that shows the model structure, including how the disease natural history is represented, the process and determinants of disease acquisition, and how the putative intervention could affect the system.

* Please provide a complete list of model parameters, including clear and precise descriptions of [the meaning of each parameter, together with the values or ranges for each, with justification or the primary source cited, and important caveats about the use of these values noted].

* For uncertainty analyses, please state the sources of uncertainties quantified and not quantified [can include parameter, data, and model structure].

* Please provide sensitivity analyses to identify which parameter values are most important in the model. Uncertainty estimates seek to derive a range of credible results on the basis of an exploration of the range of reasonable parameter values. The choice of method should be presented and justified.

* Please discuss the scientific rationale for this choice of model structure and identify points where this choice could influence conclusions drawn. Please also describe the strength of the scientific basis underlying the key model assumptions.

Methods/’Data’ – this section both defines and discusses the background to the methodology. Please ensure that the methods section serves to only clearly define the methodology used and restrict background information/discussions to the relevant sections of your manuscript. Please revise accordingly. Please define PCR at first use.

Methods/’travel testing model’ – ‘D=30 days’ what does the D depict here, please define for the reader.

Ethics statement – please define LSHTM for the reader.

As above, PLOS Medicine requests that main results are quantified with 95% CIs and p values. When reporting p values, please report as p<0.001 and where higher the exact p value as p=0.002, for example. If not reporting p values, please clearly state the reasons why not, to help facilitate transparent data reporting.

When reporting 95% CIs suggest the use of commas to separate upper and lower bounds as opposed to hyphens as these can be confused with reporting of negative values. Please check and amend throughout the main manuscript, tables, figures and supporting files where relevant. 

Results, para 5 – please clearly define the numerical values contained within parentheses for the reader

FIGURES & SUPPORTING FIGURES

Please consider avoiding the use of green and/or red to improve accessibility of your figures to those with colour blindness.

Please ensure that all figure captions clearly describe the content of the figure without the need to refer to the text.

Please ensure that all abbreviations are defined in the figure captions for the reader, including those used to report statistical information.

Figure 1 – you report credible intervals (CrI) and elsewhere confidence intervals (CI) should it be one or the other? Please clarify/revise as necessary.

Figure 3 – please define HCW, PCR, Ct, WT

Figure 5 – please clearly define the meaning of the dots and lines as well as the grey dotted line.

DISCUSSION

Please present and organize the Discussion as follows (avoiding the use of sub-headings): a short, clear summary of the article's findings; what the study adds to existing research and where and why the results may differ from previous research; strengths and limitations of the study; implications and next steps for research, clinical practice, and/or public policy; one-paragraph conclusion.

REFERENCES

Please ensure that in the bibliography, up to but no more than 6 author names are listed followed by et al., in the event that more than 6 individuals contribute to an individual study. Please ensure that journal name abbreviations are those found in the National Center for Biotechnology Information (NCBI) databases. Please see our website for other reference guidelines https://journals.plos.org/plosmedicine/s/submission-guidelines#loc-references

Comments from the reviewers:

Reviewer #1: Kucharski et al. analyse COVID testing data on persons arriving in French Polynesia, and make useful inferences on infection prevalence in other parts of the world.

Overall comments

This is a very nice and careful analysis of a complex dataset. There are some limitations, and one general question I would have is about the public health value of this data to the people paying for this data (possibly the taxpayers in French Polynesia?), since it relates to infection prevalence in other locations. Nevertheless, countries should be willing to donate resources for the greater international good. In addition, I don't think it is too plausible that COVID policies in France or the United States would be based on surveillance data on travelers into French Polynesia. France and the United States should be capable of collecting data on infection prevalence in their own locations. Notwithstanding these issues, there is clearly incredible scientific value of the data, which were generated anyway because of on-arrival testing of all arrivals. I agree with authors that a global network of sentinel locations collecting this type of data would be a fantastic resource to allow inferences on infection prevalence globally, and WHO should certainly consider this in the next pandemic. However for surveillance purposes, the ideal data would be on-arrival testing and reporting of pre-departure tests, because the shape of the "hole" in Figure 1D is absolutely critical to all of authors' inferences.

Perhaps worthy of mention is that the travel measures in French Polynesia between February and August 2021 was part of a successful effort to keep COVID out of the community (correct, from Figure 3C?) by combining it with on-arrival quarantine as well as other control measures? Perhaps the paper could be expanded with a brief mention of this, because obviously on-arrival COVID testing will only affect COVID control if combined with (1) minimization in daily arrivals and (2) strict management of those that do arrive and (3) domestic outbreak control measures. Without these other three components it is unlikely that arrival testing will have any impact.

One other final comment/concern is the complexity of methodological description used for presentation to a general audience. Authors might consider illustrating the methods with a flow diagram or schematic to show the concepts involved, with formulas left to the appendix. 

Major comments

1. Have authors assumed that PCR sensitivity in French Polynesia is equal to PCR sensitivity in the pre-departure tests (mainly in France and the US)? I would question that assumption. PCR sensitivity can vary substantially between laboratories. Figure 1D has a chunk out of it assuming that pre-departure tests have a particular sensitivity, but authors have basically no information on that. Authors also assume constant PCR sensitivity over time for different variants which is somewhat unlikely.

2. In third paragraph of results, at the end, there is an argument about the proportion of infections that would be picked up by delayed testing but the point of this analysis is unclear. Other studies have already shown that unless imported infections are reduced to an absolute trickle, COVID will spread in the community (unless the community is already in lockdown). Delayed on-arrival COVID testing may or may not have an advantage over immediate on-arrival testing but there is a far bigger story to look at there, than just the proportion of infections picked up. This paper seems to be about the inferences that can be drawn on prevalence in France and the US from the arrival testing, not about the value of the arrival testing for the control of COVID in French Polynesia (see comments above).

3. End of first paragraph of results "Our approach could therefore generate reliable estimates for simulated departure prevalence dynamics…" that's good news that your model can recover input parameters in a simulation, but reliability mainly depends on sample size, accuracy would be a more important metric to mention here, particularly in simulations where the pre-departure PCR sensitivity is misspecified in your model. At a guess, the inference on trends in prevalence would be more accurate than the inference on exact prevalence values at a given time, provided that pre-departure PCR sensitivity remains fairly constant over time even if misspecified in the model?

4. For Figure 4, how did you know where travelers came from? Was this just based on where the inbound flight had taken off from, or was it based on some type of health declaration? Some passengers may have connected onto the inbound flights and actually come from other locations? Ref 23 seems to be referring to tourists but many arriving persons would be returning residents? And ref 23 may not survive in the future, it might be better to extract the raw data you used from that website and include it as an appendix file.

5. The end of the fourth paragraph provides UK data for context but that would seem to be better suited to the Discussion than the Results section. I don't think you have any directly comparable data on prevalence in France or the US for the studied periods, which means there is not really a way to validate your results. That does reveal the potential importance of your inferences - if valid - to reveal something that otherwise is not known. But it would be preferable if you had at least some data for France or the United States to triangulate. Comparison of your PCR results with serology is not particularly convincing in this case, unless you can show perhaps how ONS PCR data closely match with ONS seropositivity as well?

Reviewer #2: In this paper, the authors describe results from a SARS-CoV-2 traveler testing program run in French Polynesia between July 2020 and March 2022. As the authors highlight, the use of traveler testing data can potentially provide a less biased estimate of local infection prevalence, serve as a leading indicator for changes in prevalence, and also contain information on the prevalence of countries where travelers originate from. The authors demonstrate an approach for analyzing arrival/departure testing data, accounting for potential biases associated with epidemic stage, e.g., rising, falling steady, and when an individual was tested wrt to travel, e.g., how far in advance of travel or how long after arriving. Additionally, they provide estimates of SARS-CoV-2 prevalence in French Polynesia, France, and the United States. Given the recent attention associated with airport testing, the work here is quite timely. However, I do have a few questions/comments, which I hope the authors find constructive.

1. Do we know how infection rates among travelers is related to infection rates in the entire population? Given that there are socio-economic biases between travelers and non-travelers and socio-economic biases in infection rates, it seems as though some correction is needed because travelers are not an iid sample from the entire population.

2. Additionally, do we know if travelers are more generally at risk of getting infected? You might imagine that the distribution of social contacts for someone who is a traveler is higher than for someone that isn't, so, all else equal, their risk of getting infected would be higher.

3. Could either points 1 or 2 above be related to why you find that traveler positivity is a leading indicator of country-wide prevalence? I could also imagine that simply having a less biased estimate would provide a leading indicator as well. 

4. Related to the above points, how do your estimates of prevalence compare to hospitalizations, test positivity, and wastewater positivity (to the extent it's available) in the various countries? Given that we expect the relationship between true prevalence and test positivity to shift during the pandemic, I'd expect to see a similar shift in the relationship between test positivity and your estimate of prevalence from travel testing. 

5. Many individuals also believe that wastewater surveillance can provide an unbiased, real-time estimate of prevalence. I would encourage the authors to add a brief discussion of how wastewater surveillance and traveler testing might complement each other. For example, wastewater surveillance also tends to be a leading indicator, but that's almost certainly due to biases in clinical testing wrt to timing of infection. As a result, I could imagine that traveler testing might prove to be a true leading indicator. 

6. In the "Prevalence Model" section, the authors provide an estimate of the mean duration of positivity (8.6 days). Was this number obtain empirically? If so, it's not clear how it estimated. If it's not an empirical estimate, where do you obtain this value? 

7. Related to the above, is there data on whether infection duration varied across variants? The reported serial interval for Omicron is between 2 and 4 days (although Abbott et al. report a generation time of 1.5 - 3.2 days, which would meant the serial interval would likely be narrower).

Abbott, S., Sherratt, K., Gerstung, M., & Funk, S. (2022). Estimation of the test to test distribution as a proxy for generation interval distribution for the Omicron variant in England. medRxiv.

Kim, D., Ali, S. T., Kim, S., Jo, J., Lim, J. S., Lee, S., & Ryu, S. (2022). Estimation of serial interval and reproduction number to quantify the transmissibility of SARS-CoV-2 omicron variant in South Korea. Viruses, 14(3), 533.

Ito, K., Piantham, C., & Nishiura, H. (2022). Estimating relative generation times and relative reproduction numbers of Omicron BA. 1 and BA. 2 with respect to Delta in Denmark. medRxiv.

Reviewer #3: 

This paper is concerned with a potentially useful method for surveillance of covid prevalence using traveller screening data. While the idea and development are interesting, there is a large number of methodological issues with the paper. Specific comments are given below.

The authors will need to be scientifically honest about the quantification of uncertainty of the proposed approach. At a somewhat simplistic level they suggest that their data, which contain information from about a thousand cases, can reconstruct and accurately estimate the epidemic dynamics of the USA and France. If this method is to be used in practice one would need a precise estimate of the associated uncertainty at the very least.

This task also involves estimating the uncertainty of functions of probabilities and explicit discussions on using the bootstrap or the delta method should be given.

In addition to the variance of those probabilities, potential bias issues should be discussed in detail, especially since some of the probabilities involved are small in which case the associated MLEs/sample proportions can be highly unstable. For such cases, methods like the Firth correction or Bayesian approaches give more accurate estimates (in terms of mean square error) and could be considered in this paper.

In addition to the technical details of the bias and variance properties of the estimates used, there is a subtle notion related to the aim of the paper. Are the travellers used for screening a truly representative sample of the population at the country of origin? And if not does this really matter? 

It would be desirable to disentangle the two and offer a fair account of what the paper does and does not achieve with the proposed approach.

Please discuss reliable ways to validate these estimates, including against independent data, possibly from seroprevalence studies. The "bootstrap prediction intervals" discussed at the end of "prevalence model" are in-sample estimates and therefore conservative.

Do the authors account for the effects of false positives/negatives and the associated uncertainty?

Please discuss the availability of the various data sources, including the cohort study of self-tested healthcare workers.

I think the authors implicitly assume in their calculations that the probability of infection during travel is negligible, please discuss.

Would working with R_t estimates offer an alternative or complementary approach to the proposed technique? In some ways it would settle some of the issues like automatically adjusting for a growing/declining epidemic in the country of origin.

When discussing the results (like the 60/40 contribution in figure 3) it would be useful to try and disentangle the relative importance of assumptions and data. Some of those results heavily depend upon the assumption made so some clarification would help.

A similar idea, of exploring traveller screening data for surveillance is explored in (Bastani et al. Nature 599, p.108-113, 2021) where additional covariate information is utilised. The authors should discuss their work in the context of the relevant literature.

Page numbering would be useful in the review process, please add them.

[LINK]

---

## [Decision Letter · Decision Letter 2]

10 Aug 2023

Dear Dr. Kucharski,

Thank you very much for re-submitting your manuscript "Real-time surveillance of international SARS-CoV-2 prevalence using systematic traveller arrival screening: a retrospective study" (PMEDICINE-D-22-04017R2) for review by PLOS Medicine.

I have discussed the paper with my colleagues and the academic editor and it was also seen again by xxx reviewers. I am pleased to say that provided the remaining editorial and production issues are dealt with we are planning to accept the paper for publication in the journal.

[LINK]

We look forward to receiving the revised manuscript by Aug 17 2023 11:59PM.   

Sincerely,

Philippa Dodd, MBBS MRCP PhD

Senior Editor 

PLOS Medicine

plosmedicine.org

Requests from Editors:

GENERAL

Thank you for your detailed and considered responses to previous editor and comments. Please see below for further comments that we require you address prior to publication.

TITLE

Please replace ‘a retrospective…’ with ‘an observational…’

AUTHOR SUMMARY

Lines 63 onwards suggest removing the CIs to improve accessibility to the non-scientific reader.

Line 72 – this statement is rather vague and doesn’t really tell us anything specific. Suggest removing.

METHODS

Line 153 - please define PCR at first use in the main manuscript.

FIGURES

If this is not possible to begin axes of graphs at zero, please show a break in the axis.

Figure 4 caption – please define ‘GAM’ for the reader.

DISCUSSION

Line 354 – suggest ‘using arrival data from French Polynesia between July 2020 and March 2022’ instead.

REFERENCES

Please ensure all web references include an accessed date.

SUPPORTING INFORMATION

Please cite your Supporting Information as outlined here: https://journals.plos.org/plosmedicine/s/supporting-information

S1 Figure – please define ‘d’ (days) in the caption for the reader, please provide a legend what the different colors and letter refer to.

S2 Figure – please define ‘FP’ in the caption for the reader, please provide a legend what the different colors and letter refer to.

Supplement – please ensure the reference format follows our guidance detailed here https://journals.plos.org/plosmedicine/s/submission-guidelines#loc-references

Please ensure all web references include an accessed date.

SOCIAL MEDIA

To help us extend the reach of your research, please detail any Twitter handles you wish to be included when we tweet this paper (including your own, your coauthors’, your institution, funder, or lab) in the manuscript submission form when you re-submit your manuscript.

COMMENTS FROM THE ACADEMIC EDITOR

The authors have done a comprehensive job in responding to all editorial and reviewer comments. The figure and legends of figures can use clarification

Comments from Reviewers:

Reviewer #1: I am satisfied with previous responses and corresponding revisions, and I have no further comments

Reviewer #3: The authors have revised their paper and this is now a substantially improved manuscript.

The response to my comment about validating the paper's estimates was cut but the overall response was satisfactory so I have no further comments.

[LINK]

---

## [Editor Report · Decision Letter 3]

22 Aug 2023

Dear Dr Kucharski, 

On behalf of my colleagues and the Academic Editor, Dr. Amitabh Suthar, I am pleased to inform you that we have agreed to publish your manuscript "Real-time surveillance of international SARS-CoV-2 prevalence using systematic traveller arrival screening: an observational study" (PMEDICINE-D-22-04017R3) in PLOS Medicine.

PRESS

Best wishes, 

Philippa Dodd, MBBS MRCP PhD 

PLOS Medicine